# PivotFEC: Enhancing Few-shot Factual Error Correction with a Pivot Task Approach using Large Language Models

**Xingwei He[1], A-Long Jin[1], Jun Ma[1], Yuan Yuan[2,3,4,†], Siu Ming Yiu[1,†]**

[1]The University of Hong Kong, Hong Kong, China
[2]School of Computer Science and Engineering, Beihang University, Beijing, China
[3]State Key Laboratory of Software, Development Environment, [4]Zhongguancun Laboratory
hexingwei15@gmail.com, ajin@eee.hku.hk, junma@hku.hk,
yuan21@buaa.edu.cn, smyiu@cs.hku.hk

## Abstract

Factual Error Correction (FEC) aims to rectify false claims by making minimal revisions to align them more accurately with supporting evidence. However, the lack of datasets containing false claims and their corresponding corrections has impeded progress in this field. Existing distantly supervised models typically employ the mask-then-correct paradigm, where a masker identifies problematic spans in false claims, followed by a corrector to predict the masked portions. Unfortunately, accurately identifying errors in claims is challenging, leading to issues like over-erasure and incorrect masking. To overcome these challenges, we present PivotFEC, a method that enhances few-shot FEC with a pivot task approach using large language models (LLMs). Specifically, we introduce a pivot task called factual error injection, which leverages LLMs (e.g., ChatGPT) to intentionally generate text containing factual errors under few-shot settings; then, the generated text with factual errors can be used to train the FEC corrector. Our experiments on a public dataset demonstrate the effectiveness of PivotFEC in two significant ways: Firstly, it improves the widely-adopted SARI metrics by 11.3 compared to the best-performing distantly supervised methods. Secondly, it outperforms its few-shot counterpart (i.e., LLMs are directly used to solve FEC) by 7.9 points in SARI, validating the efficacy of our proposed pivot task.

## 1 Introduction

ChatGPT is an artificial intelligence chatbot released by OpenAI on November 30, 2022 and built upon on the company's Generative Pre-trained Transformer (GPT) series of large language models (LLMs) (Ouyang et al., 2022; OpenAI, 2023). Since its launch, ChatGPT has garnered significant global attention due to its comprehensive and eloquent responses across various knowledge domains. Within just two months, by January 2023,

---

†Corresponding authors.

it has amassed over 100 million users. However, one drawback of ChatGPT is its tendency to generate text that is nonsensical, or unfaithful to the provided source input, referred to as hallucination (Maynez et al., 2020; Raunak et al., 2021). To address this issue, the research community has dedicated efforts to the development of *factual error correction* (**FEC**), aiming to rectify false claims with minimal modifications to make them better supported by the given evidence. Consequently, research focused on this task plays a crucial role in mitigating the problem of hallucinations in LLMs.

The most straightforward way to develop FEC systems is by fine-tuning pre-trained models, such as BART (Lewis et al., 2020a) and T5 (Raffel et al., 2020), on parallel data, consisting of false claims along with their corresponding corrections. Nevertheless, the availability of such paired data is restricted due to the tremendous labor and time required for human annotations.

To overcome the data scarcity, researchers (Shah et al., 2020; Thorne and Vlachos, 2021; Chen et al., 2023) make use of distant supervision from the fact verification dataset, FEVER (Thorne et al., 2018). FEVER is a large resource consisting of claims paired with evidence from Wikipedia, where each instance is labeled as either SUPPORTED or REFUTED based on whether the claim is supported or refuted by the corresponding evidence. Existing distantly supervised models typically follow the mask-then-correct approach (Shah et al., 2020; Thorne and Vlachos, 2021). Concretely, the fact verification classifier (FVC), trained on FEVER, acts as the masker, designed to find problematic spans within false claims. The token-level explanations (Ribeiro et al., 2016; Chen et al., 2017) of FVC are usually exploited as masks. The corrector is trained on the SUPPORTED data from FEVER, with the objective of restoring/generating the original/correct claim based on the masked claim and evidence, during training/inference. Furthermore,

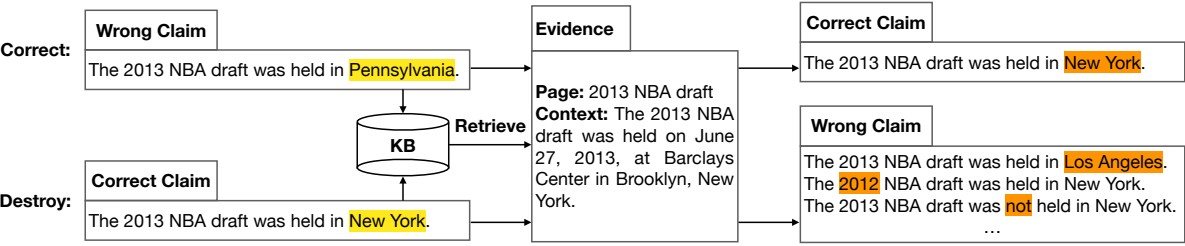

Figure 1: The comparison between factual error correction and factual error injection.

Chen et al. (2023) propose using the mask-then-correct method to iteratively refine the false claims instead of relying on a single pass. However, accurately identifying factual errors using FVC is non-trivial. This limitation often leads to over-erasure and incorrect masking issues, becoming a bottleneck that restricts the performance of FEC models.

To bypass these issues, we propose to solve the FEC task by introducing a pivot task, *factual error injection* (**FEI**), aiming to generate false claims by injecting factual errors into correct claims. Our main motivation is that the FEI task is relatively easier than the FEC task. As shown in Figure 1, an FEC model is expected to precisely identify the factual error in the false claim "The 2013 NBA draft was held in *Pennsylvania*." and correct it to "The 2013 NBA draft was held in *New York*." based on the given evidence. In contrast, the FEI task allows for multiple ways to introduce factual errors into a correct claim. For example, one can replace "*New York*" with "*Los Angeles*", substitute "*2013*" with "*2012*", or even insert a negative word such as "*not*" into the correct claim. This distinction demonstrates that the FEI task encompasses a considerably larger solution space compared to FEC. By exploring this expanded solution space, we can leverage the relatively easier nature of FEI to enhance the overall performance of FEC systems.

Our second motivation stems from the fact that LLMs, such as GPT-3.5, can serve as an excellent data annotator in few-shot settings, rivaling or even surpassing the performance of crowdsourced annotators (He et al., 2023). Inspired by this, we use LLMs to solve the FEI task, specifically generating false claims for correct claims. By doing so, we obtain a sufficient amount of paired data, which will be further used to train the FEC corrector.

Our contributions are summarized as follows: (1) We propose **PivotFEC**[1], a method that uses a **Pivot** task, factual error injection, to enhance **FEC**

---

[1] Our code is available at: https://github.com/NLPCode/PivotFEC.

with LLMs in few-shot settings. (2) Compared with distantly supervised baselines, PivotFEC only requires 8 labeled samples from FECDATA, eliminating the need for labeled data, FEVER, to train the FVC. (3) Extensive experiments conducted on the FECDATA dataset demonstrate that PivotFEC outperforms distantly supervised baselines by a large margin, achieving a new state-of-the-art (SOTA) result on the test set with scores of 66.3 on SARI and 66.68 on ROUGE-2. (4) PivotFEC exhibits much better performance than its few-shot counterpart (66.3 vs. 58.43 on SARI), where LLMs are directly used to solve FEC, confirming the effectiveness and necessity of our proposed pivot task.

## 2 Problem Statement

Factual error correction aims to revise the factual errors in claim $C$ with minimal edits and generate a revised claim $C'$ based on the provided evidence $E$. $C'$ should be grammatical, supported by the evidence, and correct the factual errors in $C$.

## 3 Preliminary

In this section, we will introduce in-context learning and how to solve FEC using LLMs with in-context few-shot learning via prompting.

### 3.1 LLMs with In-context Learning

LLMs, especially ChatGPT, have demonstrated remarkable few-shot capability in various downstream tasks. Therefore, it is natural to employ LLMs for addressing the FEC task in a few-shot setting. Building upon the approach introduced by GPT-3 (Brown et al., 2020), we utilize LLMs with in-context few-shot learning through prompting to tackle FEC. Rather than fine-tuning LLMs specifically for individual tasks, we can efficiently prompt the model by providing a small set of input-output exemplars that demonstrate the task.

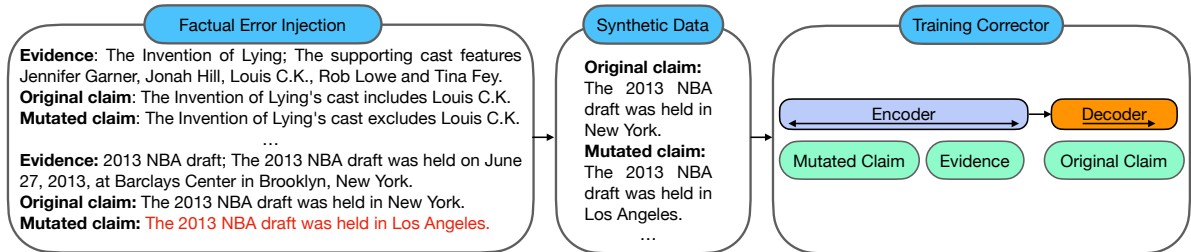

Figure 2: The PivotFEC method contains three steps: (1) prompt a LLM using FEI prompting (Red text indicates ChatGPT's output, while the rest is the input); (2) collect the generated data as the synthetic FEC data; (3) train the factual error corrector with the synthetic data. For simplicity, the process of evidence retrieval has been omitted.

## 3.2 Factual Error Correction with LLMs

To address FEC using LLMs, we begin by choosing a set of demonstrated examples. Each example comprises three elements: a gold evidence, a mutated claim, and an original claim. The objective is for LLMs to learn how to modify the mutated claim based on the provided samples. Figure 3 illustrates a simplified prompt where the LLM (in this case, ChatGPT) accurately corrects the factual error in the mutated claim by replacing "*Los Angeles*" with "*New York*." For the full prompt of the FEC task, please refer to Table 6 in Appendix C.

## 4 Approach

In this section, we will first provide an overview of PivotFEC in §4.1. As illustrated in Figure 2, our method comprises three main steps. We will begin by introducing the FEI task and demonstrating the utilization of LLMs to address the FEI task in §4.2. Next, we will present the process of gathering synthetic paired data for the FEC task in §4.3. Finally, we will demonstrate the training of the corrector using the synthetic data in §4.4.

## 4.1 Overview

The main limitation in developing FEC systems is the scarcity of paired data comprising correct claims and their corresponding false claims. To mitigate this limitation, previous studies (Shah et al., 2020; Thorne and Vlachos, 2021; Chen et al., 2023) follow the mask-then-correct method, with the assumption that there are sufficient human annotated fact verification data (i.e., FEVER), which is used to train the FVC. They train the corrector by masking certain portions of correct claims and then recovering them. Therefore, during testing, it becomes necessary to identify factual errors within false claims and mask these errors with the FVC before using the corrector to revise them. Con-

sequently, previous approaches suffer from issues such as over-erasure and incorrect masking.

Our primary motivation is to generate false claims by injecting factual errors into correct claims. This allows us to obtain FEC data consisting of correct claims paired with their corresponding false claims, which can be directly used for training the FEC corrector. To achieve this goal, we introduce the pivot task, factual error injection (FEI), for FEC, and then employ LLMs to address the FEI task using a few-shot in-context learning approach. Compared to the previous mask-then-correct method, our approach eliminates the need to mask factual errors before correction, thus avoiding the over-erasure and incorrect masking issues. Moreover, our approach does not depend on labeled fact verification data. Instead, we only require correct claims and a few labeled FEC samples.

## 4.2 Factual Error Injection with LLMs

To address FEC, LLMs are expected to identify factual errors and correct them. As previously analyzed, FEI requires LLMs to introduce factual errors into correct claims and has a significantly larger solution space than FEC (see Figure 1). Therefore, we assume that FEI is comparatively easier for LLMs than FEC. This is why we intro-

---

**Factual Error Correction**

**Evidence**: The Invention of Lying; The supporting cast features Jennifer Garner, Jonah Hill, Louis C.K., Rob Lowe and Tina Fey.
**Mutated claim**: The Invention of Lying's cast excludes Louis C.K.
**Original claim**: The Invention of Lying's cast includes Louis C.K.
...
**Evidence:** 2013 NBA draft; The 2013 NBA draft was held on June 27, 2013, at Barclays Center in Brooklyn, New York.
**Mutated claim:** The 2013 NBA draft was held in Los Angeles.
**Original claim:** The 2013 NBA draft was held in New York.

Figure 3: Factual error correction with in-context learning using ChatGPT. Text in red color denotes the output of ChatGPT, while the remaining parts are the input.

duce this pivot task for FEC.

Similar to FEC, we also employ LLMs to tackle FEI using the few-shot in-context learning approach. For fair comparisons, we utilize the same demonstrated exemplars as used in few-shot FEC, with the only difference being the order of the original claim and mutated claim. The left portion of Figure 2 illustrates a simplified prompt for FEI, where the LLM (specifically ChatGPT) injects a factual error by substituting "*New York*" with "*Los Angeles*." For the complete prompt of the FEI task, please refer to Table 7 in Appendix D.

### 4.3 Creating Synthetic Data for FEC

We assume that correct claims are readily available. For each correct claim $C^t$, we use LLMs to inject factual errors into $C^t$ via in-context learning with the prompt in Table 7. The generated claim is referred to as $C^f$. By doing so, we collect the synthetic data $\mathcal{D} = \{(C_1^t, C_1^f), \ldots, (C_i^t, C_i^f)\}$ for FEC, where $C_i^t$ and $C_i^f$ denote the $i$-th correct claim and the corresponding false (i.e., generated) claim, respectively.

### 4.4 Training FEC Corrector

After obtaining the synthetic data $\mathcal{D}$, we acquire the FEC corrector by fine-tuning pre-trained language models, such as BART or T5 on this data. To be concrete, we concatenate the false claim $C^f$ and the gold evidence or retrieved evidence, and directly input them into the encoder (refer to the right part of Figure 2 for the input format). For more detailed information on obtaining evidence for the false claim, please refer to Appendix B.1. During training, we optimize the corrector by maximizing the conditional probability of $C^t$:

$$P(C_i^t|C_i^f, E_i; \theta) = \prod_{n=1}^{N} p(C_{i,n}^t|C_{i,j<n}^t, C_i^f, E_i; \theta),$$

where $\theta$ represents the parameters of the corrector, $E_i$ denotes the corresponding evidence, and $C_{i,j<n}^t$ refers to the sub-sequence preceding $C_{i,n}^t$.

## 5 Experiment

### 5.1 Experimental Setups

**Dataset.** Following previous work, we evaluate our model on the evidence-based FEC dataset (FECDATA) (Thorne and Vlachos, 2021), created based on the large fact verification dataset, FEVER (Thorne et al., 2018). The construction of the

| Label | Train | Valid | Test |
|---|---|---|---|
| SUPPORTED | 37,961 | 1,477 | 1,593 |
| REFUTED | 20,075 | 2,091 | 2,289 |
| Total | 58,036 | 3,568 | 3,882 |

Table 1: The basic statistics of FECDATA with the number of data instances for each split and label.

FEVER dataset involves two main steps: claim generation and claim labeling. In the claim generation phase, annotators extract the original claims (i.e., correct claims) from Wikipedia, and then use six types of mutation: paraphrasing, negation, substitution of an entity/relation with a similar/dissimilar one, and making the claim more general/specific to generate mutated claims for original claims. In the claim labeling phase, annotators classify claims as SUPPORTED, REFUTED or NOTENOUGHINFO based on whether the claim is supported, refuted or unverifiable by the given evidence.

FECDATA extracts the SUPPORTED and REFUTED data instances from FEVER, and uses the original claims and mutated claims as the paired data. Table 1 shows the basic statistics of this dataset. To gain further insights, we provide Figure 6 in Appendix A, displaying the distribution of mutation types for the REFUTED claims.

**Evaluation Metrics.** For automatic evaluation, we resort to SARI (Xu et al., 2016)[2] and ROUGE-2 (Lin, 2004)[3] metrics. The SARI metric explicitly assesses the goodness of words in the revised claim that are added, deleted and kept by FEC models from the source (mutated claim), compared with the referenced ground truth (original claim). We report the n-gram F1 score for "keep" operations (**Keep**), the n-gram precision score for "delete" operations (**Delete**), the n-gram F1 score for "add" operations (**Add**), and the average of these three scores (**Final**). **ROUGE-2** measures the surface-level similarities between revised claims and reference claims. The SARI Final score serves as the primary evaluation metric due to its strong positive correlation with manual evaluation, as indicated by Thorne and Vlachos (2021)'s statistical findings.

**Baselines.** We consider three types of baselines:
**Fully Supervised Baselines** estimate the ceiling performance of FEC models, under the assump-

---

[2]The evaluation code for SARI is available at: https://huggingface.co/spaces/evaluate-metric/sari.

[3]The evaluation code for ROUGE is available at: https://huggingface.co/spaces/evaluate-metric/rouge.

tion that a substantial amount of data is accessible. For this purpose, we fine-tune **BART-base** and **T5-base** on FECDATA, where the encoder takes the false claim and corresponding evidence as inputs, while the decoder generates the revised claim.

**Distantly Supervised Baselines** adopt the '*mask-then-correct*' pipeline, consisting of a masker and a corrector. The masker can take various forms, such as the token-level explanations (Ribeiro et al., 2016; Chen et al., 2017) of a fact verification classifier (FVC), random masking, or heuristic masking. The FVC is initialized with BERT-base (Devlin et al., 2019) or RoBERTa-large (Liu et al., 2019), and trained on FEVER. On the other hand, the corrector is trained exclusively on the SUPPORTED data instances from FEVER. (1) Dual encoder pointer network (**DEPN**) (Shah et al., 2020) utilizes an FVC to predict masked words and subsequently generates a revised claim using the dual encoder pointer generator with the copy mechanism (See et al., 2017). (2) T5 Masker-Corrector (**T5MC**) (Thorne and Vlachos, 2021) differs from DEPN in two main ways: (a) The corrector (i.e., generator) is based on T5-base. (b) It randomly masks words in the input claim during training, but during inference, heuristic masking is employed, where words do not present in the evidence are masked. (3) **T5MC-MLM** differs from T5MC in that it uses the masked language model BERT as the masker during inference. (4) **T5MC-V** is a variant of T5MC, using FVC as the masker to predict the masked tokens. (5) **VENCE** (Chen et al., 2023) iteratively executes steps of mask-then-correct over the claim to make it supported by the evidence.

**Rule-based Baselines** first generate synthetic paired data, and then train factual error correctors on them. Rule-based methods are employed to construct the inconsistent summaries with the aim of improving the faithfulness in abstractive summarization (Cao et al., 2020; Cao and Wang, 2021). We begin by utilizing spaCy[4], a free open-source library for NLP, to recognize name entities in correct claims, and then implement two rule-based baselines for factual error correction: (1) The first rule-based method creates false claims by swapping named entities from the correct claims with alternative entities of the same entity type randomly chosen from the training dataset. This method is referred to as **SwapEntity**. (2) The second rule-based baseline resorts to the '*mask-then-fill*' pipeline to

create false claims. In this approach, named entities within the correct claims are substituted with [MASK] tokens. The masked claims are then processed through the BART-base model to generate new claims by filling in the [MASK] tokens. These newly generated claims are considered as false claims. This method is termed **MaskEntity**.

**Few-shot Baselines** contains two models: **8-shot T5-base** fine-tunes T5-base using 8 data examples. **8-shot ChatGPT** revises false claims by prompting ChatGPT with 8 demonstrated examples. For fair comparisons, the few-shot baselines and our PivotFEC use the same set of examples.

**Implementation Details.** Our implementation details are shown in Appendix B.

## 5.2 Experimental Results

The main experimental results on the FECDATA test set in Table 2 reveal the following key findings: **LLMs exhibit a remarkable few-shot ability for FEC.** Directly fine-tuning T5 on the 8 labeled data instances (i.e., 8-shot T5-base) does not bring any improvement over previous distantly supervised baselines, such as VENCE. However, the few-shot in-context learning baseline (i.e., 8-shot ChatGPT) achieves a noteworthy SARI Final score of 58.43, surpassing VENCE (RoBERTa) by approximately 3 points. These results highlight the impressive few-shot capability of ChatGPT.

**Our proposed pivot task is highly effective for few-shot FEC.** To demonstrate the effectiveness of our proposed PivotFEC, we compare it with the 8-shot ChatGPT model. To ensure fair comparisons, both few-shot models are based on ChatGPT. As shown in Table 2, our proposed model, 8-shot PivotFEC, far exceeds its few-shot counterpart (8-shot ChatGPT) by a significant margin across all metrics. The SARI metric increased from 58.43 to 66.30 and the ROUGE-2 score increased from 49.43 to 66.68.

On the other hand, PivotFEC also notably outperforms the rule-based baselines. Additionally, PivotFEC achieves the peak performance with just 2,000 synthetic data instances for training the corrector, while rule-based methods require 10,000 synthetic data instances to reach their peak performance when training the correctors. These results can be attributed to the enhanced quality of false claims produced by ChatGPT. Rule-based methods, by contrast, often produce false claims with suboptimal quality in two key aspects: (1) Grammatical

---
[4] https://spacy.io/

Table 2:

| Models | FVC | Retrieved Evidence | | | | | | Gold Evidence | | | | |
| | | SARI Score | | | | RG-2 | | SARI Score | | | | RG-2 |
| | | Keep | Delete | Add | Final | | | Keep | Delete | Add | Final | |
| **Fully Supervised Baselines** | | | | | | | | | | | | |
| Supervised BART-base* | - | 70.75 | 65.65 | 38.88 | 58.43 | 59.99 | | 73.51 | 69.27 | 47.30 | 63.36 | 64.00 |
| Supervised T5-base* | - | 85.40 | 88.92 | 48.40 | 74.24 | 73.50 | | 88.56 | 91.40 | 58.38 | 79.45 | 78.04 |
| **Distantly Supervised Baselines** | | | | | | | | | | | | |
| DEPN (Shah et al., 2020)‡ | Bb | 34.5 | 48.1 | 1.7 | 28.1 | 34.8 | | 45.2 | 56.9 | 3.9 | 35.3 | - |
| T5MC (Thorne and Vlachos, 2021)† | - | 65.2 | 62.7 | 15.5 | 47.8 | 50.3 | | 66.7 | 62.2 | 16.1 | 48.3 | - |
| + Enumerate† | Bb | 66.2 | 64.3 | 17.1 | 49.2 | 51.2 | | - | - | - | - | - |
| T5MC-MLM‡ | - | 56.1 | 52.9 | 7.8 | 38.9 | 42.7 | | - | - | - | - | - |
| T5MC-V (Thorne and Vlachos, 2021)† | Bb | 61.1 | 54.3 | 19.4 | 44.9 | 42.0 | | 61.8 | 62.2 | 10.2 | 44.7 | - |
| + Enumerate† | Bb | 63.0 | 55.7 | 24.1 | 47.6 | 45.5 | | - | - | - | - | - |
| VENCE (Chen et al., 2023)† | Bb | 66.0 | 60.1 | 34.8 | 53.6 | 57.7 | | 67.5 | 61.5 | 34.6 | 54.5 | - |
| | Rl | 67.1 | 61.9 | 36.0 | 55.0 | 59.1 | | - | - | - | - | - |
| **Rule-based Baselines** | | | | | | | | | | | | |
| SwapEntity* | - | 67.95 | 94.57 | 16.62 | 59.71 | 62.06 | | 70.32 | 97.72 | 22.25 | 63.43 | 64.88 |
| MaskEntity* | - | 70.31 | 94.37 | 19.04 | 61.24 | 63.49 | | 73.91 | 94.92 | 27.81 | 65.55 | 66.45 |
| **Few-shot Baselines** | | | | | | | | | | | | |
| 8-shot T5-base* | - | 61.75 | 85.23 | 8.70 | 51.89 | 49.83 | | 63.50 | 82.44 | 13.56 | 53.17 | 51.38 |
| 8-shot ChatGPT* | - | 72.09 | 75.92 | 27.29 | 58.43 | 49.43 | | 79.98 | 81.61 | 38.81 | 66.80 | 60.72 |
| **Few-shot (Our Method)** | | | | | | | | | | | | |
| 8-shot PivotFEC (ChatGPT)* | - | 76.51 | 92.61 | 29.78 | **66.30** | **66.68** | | 79.62 | 93.82 | 39.21 | **70.89** | **70.41** |

Table 2: Automatic evaluation results (%) of our model and baselines with retrieved evidence or ground truth evidence on the FECDATA test set. Results marked with †, ‡, and ∗ are from VENCE (Chen et al., 2023), T5MC-V (Thorne and Vlachos, 2021) and our reproduction, respectively. Bb and Rl denote BERT-base and RoBERTa-large. Enumerate refers to using the FVC model to rank 20 generated claims and select the best one. underline indicates the best model and **bold** indicates the second best. RG-2 refers to ROUGE-2.

errors might be present in generated false claims. (2) False claims generated by rule-based methods might deviate from the original topics present in correct claims. With ChatGPT's remarkable in-context learning capabilities, injecting factual errors into correct claims hardly introduces grammatical errors or deviates from the original topics.

These compelling improvements establish a new SOTA result and provide strong evidence for the effectiveness of the pivot task in enhancing the performance of FEC.

**PivotFEC lags behind supervised baselines.** While PivotFEC outperforms distantly supervised and few-shot models, there still exists a significant performance gap compared to supervised methods. For example, the supervised T5-base achieves a score of 74.24 on SARI Final, whereas PivotFEC only scores 66.30, indicating that there is ample room for further improvement.

**The retrieved evidence is inadequate compared with gold evidence.** To further explore the ceiling performance of our method, we conduct experiments using gold evidence as well. The results reveal that when using gold evidence, PivotFEC improves the SARI Final score by approximately 4 points. This demonstrates the inadequacy of retrieved evidence, which aligns with previous findings (Thorne and Vlachos, 2021). As our work mainly focuses on improving FEC through the introduction of the pivot task, we defer the improvement of evidence retrieval to future work.

### 5.3 More Analysis and Discussion

The experiments conducted in this section utilize ChatGPT with 8-shot in-context learning and retrieved evidence, if there is no particular statement.

**Effect of the Number of Synthetic Data.** To show the effect of synthetic data generated with FEI on PivotFEC, we first generate synthetic data for FEC with 8-shot PivotFEC (ChatGPT), and then fine-tune T5-base on varied numbers of synthetic data instances. For comparison, we also evaluate the performance of T5-base trained on the gold data (FECDATA). As shown in Figure 4 (a), when the data size does not exceed 1k, the performance of 8-shot PivotFEC increases linearly with the increase of data, even matching the performance of T5-base trained on the gold data. Nevertheless, our

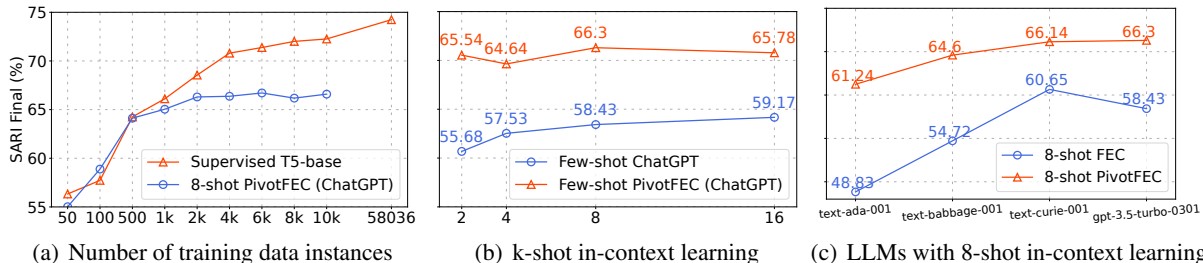

(a) Number of training data instances    (b) k-shot in-context learning    (c) LLMs with 8-shot in-context learning

Figure 4: Subfigure (a) shows the performance on the test set for T5-base trained with different numbers of gold or generated data instances. Subfigure (b) shows the performance on the test set for different few-shot in-context learning. Subfigure (c) shows the performance on the test set for different LLMs with 8-shot in-context learning.

model's performance plateaus once the data size reaches 2k. In contrast, T5-base trained on gold data continues to improve. Even when using all FECDATA training data, its performance does not reach its peak. This observation suggests that the generated data contains noise compared to the gold data, limiting the upper performance of PivotFEC.

**Effect of the Number of In-Context Examples.** Table 2 demonstrates the superiority of PivotFEC over the few-shot FEC with ChatGPT under the 8-shot setting. To further validate the effectiveness of PivotFEC, we compare its performance with the few-shot FEC using ChatGPT with varying numbers of in-context examples. As depicted in Figure 4 (b), PivotFEC exhibits a notable improvement of approximately 7 to 10 points on the SARI Final score compared to the few-shot FEC model at different shots. When utilizing 8 demonstrated examples, our model reaches a plateau.

**Effect of Different LLMs.** To further emphasize the advantages of our method, we compare it with 8-shot FEC across different LLMs, including three InstructGPT models (text-ada-001, text-babbage-001, and text-curie-001) and ChatGPT (gpt-3.5-turbo-0301). Figure 4 (c) illustrates our method consistently outperforms the few-shot FEC baseline across different LLMs. Moreover, both our method and the baseline exhibit noticeable performance improvements as the model parameters increase. However, even with small models, our method still performs exceptionally well. For example, the smallest model falls only around 5 points behind the largest model. In contrast, the baseline's performance is heavily influenced by the choice of model, particularly evident in text-ada-001, which experiences a decrease of approximately 12 points compared to the larger model, text-curie-001.

| Mutation Types | SARI Score | | | | RG-2 |
|---|---|---|---|---|---|
| | Keep | Delete | Add | Final | |
| Negate | 76.51 | 92.61 | 29.78 | **66.30** | **66.68** |
| Substitute similar | 73.72 | 92.08 | 25.16 | 63.66 | 64.97 |
| Substitute dissimilar | 69.80 | 92.47 | 17.99 | 60.09 | 61.43 |
| Specific | 71.33 | 92.52 | 22.53 | 62.13 | 64.09 |

Table 3: Evaluation results (%) of PivotFEC with prompts created using examples from different mutation types on the test set. RG-2 denotes ROUGE-2.

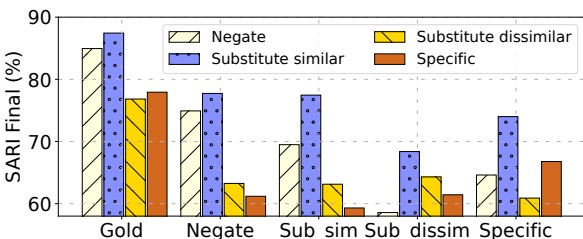

Figure 5: Results on different test cases, with the test set being divided according to mutation types. Gold denotes T5-base trained on FECDATA. Negate, Sub_sim, Sub_dissim, and Specific refer to PivotFEC with their respective mutation prompts.

**Effect of Mutation Types.** As shown in Figure 6 in Appendix A, the refuted claims mainly stem from four mutation types. Therefore, we construct four prompts, each composed of examples from the corresponding mutation. For prompts consisting of examples from *negate*, *substitute similar*, *substitute dissimilar*, and *specific* mutations, please refer to Tables 7, 8, 9, and 10. Table 3 shows that PivotFEC with *negate* prompt yields the best results, possibly because this mutation constitutes the largest portion of the test set. Additionally, we present the performance of PivotFEC on separate test cases. Figure 5 illustrates that: (1) PivotFEC performs well on a test case of the specific mutation type when using a prompt tailored to that type, and (2) the variations in PivotFEC performance, when us-

| Models | Grammar | Support | Correct |
|---|---|---|---|
| **Fully supervised models with gold evidence** | | | |
| T5-base | 100 | 89.3 | 86.7 |
| **8-shot models with retrieved evidence** | | | |
| T5-base | 83.3 | 22.0 | 5.3 |
| ChatGPT | 92.0 | 90.0 | 42.0 |
| PivotFEC (ChatGPT) | 99.3 | 65.3 | **54.7** |

Table 4: Human evaluation results (%) on the test set for the grammatical (**Grammar**), supported (**Support**) and corrected (**Correct**) scores.

ing different prompts, mainly arise from the *negate* test case.

### 5.4 Human Evaluation

We conduct a human evaluation to compare PivotFEC with the fully supervised T5-base, 8-shot T5-base and 8-shot ChatGPT models. The fully supervised T5-base model utilizes gold evidence to rectify false claims, while the others use retrieved evidence. For each model, we randomly sample 50 cases and ask three annotators[5] to assess the revised claims based on the following Boolean questions: (1) Is the revised claim grammatically correct? (2) Is the revised claim supported by evidence? (3) Has the factual error in the false claim been corrected? The final question, measuring the correction of factual errors, is the most important metric in our human evaluation. As shown in Table 4, our proposed model outperforms the few-shot baselines on the corrected metric; however, there is still a gap to reach the ceiling performance of the supervised baseline. Inter-annotator agreement measured by Fleiss' *kappa* (Fleiss, 1971) is 0.75, 0.86, and 0.81 for grammatical, supported, and corrected scores, implying substantial agreement ($> 0.6$) (Landis and Koch, 1977).

### 5.5 Samples and Analysis

Table 5 presents the revised claims generated by our approach and the baselines. From this table, we observe that the 8-shot T5-base method cannot identify errors in false claims. Similarly, 8-shot ChatGPT often struggles to precisely locate errors within false claims, and tends to simply copy content from the evidence into the modified claims rather than correct them. For example, in the first example, although 8-shot ChatGPT corrects the factual error in the original sentence, it does not

---

[5]All annotators hold Ph.D. degrees and are independent from our research.

make the minimal edits. As for the second example, 8-shot ChatGPT fails to identify the error in the false claim, resulting in a text that exhibits a low correlation with the original claim. This also explains why this method achieves a relatively high supported value but demonstrates a low corrected score during the human evaluation. Most notably, our method can accurately identify errors and make modifications based on the retrieved evidence, similar to the performance of the supervised T5 model.

## 6 Related Work

### 6.1 Grammatical Error Correction

Grammatical error correction (GEC) (Ng et al., 2014; Yuan and Briscoe, 2016; Bryant et al., 2017; Awasthi et al., 2019; Liu et al., 2021) refers to the process of identifying and rectifying grammatical errors in written text. It has practical applications in several domains, such as helping nonnative speakers enhance their writing skills, aiding language learners in improving their grammatical accuracy, and assisting professional writers in producing error-free and polished content. GEC aims to improve the accuracy and fluency of the language by fixing various grammatical errors, including missing prepositions, mismatched subject-verb agreement, misspellings, and word choice errors. In comparison, factual error correction involves correcting the factual errors instead of the grammatical errors in the given content, such as incorrect dates, names, or historical events.

### 6.2 Retrieval-Augmented Generation

Retrieval-augmented generation (Lewis et al., 2020b) combines the power of information retrieval and language generation techniques to elevate the overall quality of the generated content. For example, (He et al., 2022) use dense retrievers to retrieve relevant sentences from an external corpus for the given keywords to improve lexically constrained text generation (He and Li, 2021; He, 2021). By incorporating external knowledge, retrieval-augmented generation effectively mitigates the risk of generating inaccurate or nonsensical content. Furthermore, factual error correction is another facet of retrieval-augmented generation. It rectifies factual inaccuracies based on the evidence retrieved, thereby fitting under the broader umbrella of retrieval-augmented generation.

| | |
|---|---|
| **Gold Claim:** The SpongeBob SquarePants Movie is a film. | |
| **False Claim:** The SpongeBob SquarePants Movie is a dog. | |
| **Retrieved Evidence:** title: The SpongeBob SquarePants Movie; context: The SpongeBob SquarePants Movie is a 2004 American animated adventure comedy film based on the Nickelodeon television series SpongeBob SquarePants ... | |
| **Supervised T5-base:** The SpongeBob SquarePants Movie is a film. | ✓ |
| **8-shot T5-base:** The SpongeBob SquarePants Movie is a dog. | ✗ |
| **8-shot ChatGPT:** The SpongeBob SquarePants Movie is an animated film based on the Nickelodeon television series SpongeBob SquarePants. | ✓ |
| **8-shot PivotFEC (ChatGPT):** The SpongeBob SquarePants Movie is a film. | ✓ |
| **Gold Claim:** The Hunger Games are based on a novel. | |
| **False Claim:** The Hunger Games are based on a play. | |
| **Retrieved Evidence:** title: The Hunger Games; context: ... The Hunger Games is a trilogy of young adult dystopian novels written by American novelist Suzanne Collins ... | |
| **Supervised T5-base:** The Hunger Games are based on a novel. | ✓ |
| **8-shot T5-base:** The Hunger Games are based on a play. | ✗ |
| **8-shot ChatGPT:** The Hunger Games is a trilogy of young adult dystopian novels. | ✗ |
| **8-shot PivotFEC (ChatGPT):** The Hunger Games are based on a novel. | ✓ |

Table 5: Revised claims generated by our model and baselines based on the evidence for false claims extracted from the test set. The supervised T5-base revises false claims based on the gold evidence, while others utilize retrieved evidence. For simplicity, we do not show the gold evidence. Text in blue, red, and orange colors represents factual errors, correct modifications, and copied text, respectively.

## 6.3 Fact Verification

Fact verification, also known as fact-checking, aims to validate the accuracy of a given claim by examining the available evidence. This field of study has been extensively researched in recent years. Researchers assess the claim by analyzing both unstructured sources, such as political news (PolitiFact) (Vlachos and Riedel, 2014; Wang, 2017), Wikipedia (FEVER) (Thorne et al., 2018; Liu et al., 2020), and scientific literature (Wadden et al., 2020), as well as structured sources, including Wikipedia tables (TabFact) (Chen et al., 2020) and knowledge base (Iso et al., 2020). Fact verification seeks to determine whether a claim is supported or refuted by evidence, while factual error correction takes it a step further. Factual error correction not only involves identifying factual errors but also requires modifying them to obtain correct claims.

## 7 Conclusion

In this paper, we present PivotFEC, which introduces a pivot task, factual error injection, to improve factual error correction. Specifically, we first intentionally introduce factual errors into correct claims using LLMs under the few-shot setting. By doing so, we can obtain enough synthetic paired data for FEC, consisting of correct claims paired with their corresponding false claims, which will be used to train the FEC corrector. As a result, PivotFEC demonstrates a significant improvement over previous distantly supervised baselines, establishing a new SOTA performance on FECDATA. Furthermore, our approach significantly outperforms its few-shot counterpart, providing strong evidence for the effectiveness of the pivot task.

## 8 Limitations

There are two potential limitations to this study. Firstly, due to limited computational resources, we only assess the effectiveness of our proposed method on GPT series models. Future work should include additional experiments with other language models, such as PaLM and LLaMA. The second limitation is that the retrieved evidence may not always be relevant to the input claim, which means it may not provide useful information for correcting factual errors within the claim. As our primary focus is on enhancing factual error correction through the introduction of the pivot task, we leave the task of improving evidence retrieval for future research.

## Acknowledgements

This project is supported by National Natural Science Foundation of China (Grant No. 62202023), HKU-SCF FinTech Academy, Shenzhen-Hong Kong-Macao Science and Technology Plan Project (Category C Project: SGDX20210823103537030), and Theme-based Research Scheme of RGC, Hong Kong (T35-710/20-R). We would like to thank the anonymous reviewers for their constructive and informative feedback on this work.

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

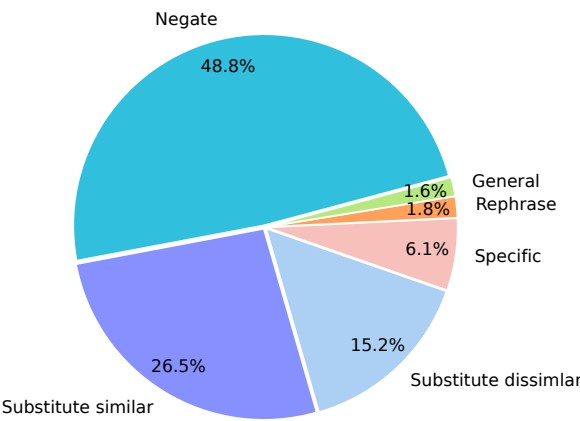

Figure 6: Distribution of mutation types for refuted claims of the test set.

## A Distribution of Mutation Types

Figure 6 presents the distribution of mutation types for revised claims of the test set.

## B Implementation Details

### B.1 Evidence Retrieval

Considering that our research does not focus on improving the retrieval model, we adopt the same retrieval process as previous studies (Thorne and Vlachos, 2021; Chen et al., 2023). The retrieval module primarily consists of two steps: First, we employ a pre-trained seq2seq model called GENRE (Cao et al., 2021) to predict the relevant Wikipedia pages for the input claim. Then, we utilize the dense passage retrieval model DPR (Karpukhin et al., 2020) to retrieve the most relevant passages from the pages predicted by GENRE.

### B.2 PivotFEC

**Synthetic data for FEC.** We generate synthetic FEC data using REFUTED data instances from FECDATA by solving the FEI task. Since we assume that correct claims are readily available, we utilize only the correct claims from the REFUTED data instances and exclude their paired false claims.

To provide clarity, we construct a total of 1296 validation data instances and 2000 training data instances. It's worth noting that increasing the amount of training data does not yield substantial improvements, as discussed in Section 5.3.

**Training and Inference.** During training, we initialize the PivotFEC corrector with the T5-base model. Following previous work (Thorne and Vlachos, 2021), we input the top-2 retrieved evidence or gold evidence paired with the false

claim into the T5 encoder, as additional evidence does not yield significant improvements. The corrector is optimized using the AdamW optimizer (Loshchilov and Hutter, 2019) with a learning rate of $4e-5$, a batch size of $64$, and a linear learning rate schedule with $10\%$ warm-up steps for $400$ steps. The learning rate is selected from the set $\{5e-6, 1e-5, 2e-5, 3e-5, 4e-5, 5e-5\}$. We set the maximum source length to $512$ and the maximum target length to $256$. During training, we evaluate the model every $50$ steps on the synthetic validation set and choose the checkpoint with the lowest negative log-likelihood (NLL) loss on the validation set.

During inference, we employ beam search decoding with a beam width of $5$ to generate revised claims for the test set.

### B.3 Supervised Models

We fine-tune the pre-trained models, BART-base and T5-base, on the FECDATA training set for $4000$ steps. We evaluate the model every $200$ steps using the FECDATA validation set. Other parameters remain consistent with those of PivotFEC, as stated in Section B.2.

To implement all models, we utilize the HuggingFace Transformers library (Wolf et al., 2019). Additionally, all experiments are conducted on 2 NVIDIA Tesla V100 GPUs with 32 GB of memory.

## C Full Few-shot Prompts for FEC

Table 6 shows the few-shot exemplars prompt of the *negate* mutation type for the FEC task. Table 6 and Table 7 use the same demonstrated exemplars with the only difference being the order of the original claim and mutated claim.

## D Full Few-shot Prompts for FEI

Tables 7, 8, 9, and 10 show the few-shot exemplars prompt of the *negate*, *substitute similar*, *substitute dissimilar* and *specific* mutation types for the FEI task, respectively.

**Evidence:** The Lion King; The story takes place in a kingdom of lions in Africa and was influenced by William Shakespeare 's Hamlet .
The Lion King; The Lion King tells the story of Simba , a young lion who is to succeed his father , Mufasa , as King of the Pride Lands ; however , after Simba 's uncle Scar murders Mufasa , Simba is manipulated into thinking he was responsible and flees into exile .
**Mutated claim:** The Lion King has nothing to do with lions.
**Original claim:** The Lion King is about lions.

**Evidence:** Indiana Jones; Henry Walton " Indiana " Jones Jr. ( also shortened to Indy ) is a fictional character and the protagonist of the Indiana Jones franchise .
Indiana Jones; George Lucas created the character in homage to the action heroes of 1930s film serials .
**Mutated claim:** Indiana Jones is real.
**Original claim:** Indiana Jones is fictional.

**Evidence:** Scott Eastwood; Scott Eastwood ( born Scott Clinton Reeves ; March 21 , 1986 ) is an American actor , model , and professional skydiver .
Scott Eastwood; He has also been the model for the fragrance Cool Water by Davidoff .
**Mutated claim:** Scott Eastwood was incapable of working as a model.
**Original claim:** Scott Eastwood worked as a model.

**Evidence:** Akshay Kumar; Kumar is also a producer and martial artist who has appeared in over a hundred Hindi films .
Akshay Kumar; Having done so , he has established himself as a leading contemporary actor of Hindi cinema .
**Mutated claim:** Akshay Kumar does not work in Hindi cinema.
**Original claim:** Akshay Kumar works in Hindi cinema.

**Evidence:** Gorillaz; Gorillaz are an English virtual band created in 1998 by musician Damon Albarn and artist Jamie Hewlett .
Virtual band; In music , a virtual band ( also called a virtual group , cartoon group , or cartoon band ) is any group whose members are not corporeal musicians , but animated characters .
**Mutated claim:** Gorillaz is a German live band.
**Original claim:** Gorillaz is a British virtual band.

**Evidence:** Grant Gustin; Thomas Grant Gustin ( born January 14 , 1990 ) is an American actor , singer , and dancer .
Grant Gustin; He is known for his role as Barry Allen / the Flash ( based on the DC Comics character of the same name ) on the CW series The Flash and Arrow , both in the Arrowverse television franchise , and his role as Sebastian Smythe on the Fox series Glee .
**Mutated claim:** Grant Gustin is only a writer.
**Original claim:** Grant Gustin is a singer.

**Evidence:** RB Leipzig; RasenBallsport Leipzig e.V. , commonly known as RB Leipzig , is a German association football club based in Leipzig , Saxony .
Football in Germany; Football is the most popular sport in Germany .
**Mutated claim:** RB Leipzig plays the least popular German sport.
**Original claim:** RB Leipzig plays the most popular German sport.

**Evidence:** One World Trade Center; One World Trade Center ( also known as 1 World Trade Center , 1 WTC or Freedom Tower ) is the main building of the rebuilt World Trade Center complex in Lower Manhattan , New York City .
World Trade Center (2001–present); The original World Trade Center featured the landmark Twin Towers , which opened in 1973 , and were the tallest buildings in the world at their completion .
**Mutated claim:** One World Trade Center opened in 1876.
**Original claim:** One World Trade Center opened in 2014.

Table 6: Few-shot exemplars prompt of the *negate* mutation type for the FEC task.

**Evidence:** The Lion King; The story takes place in a kingdom of lions in Africa and was influenced by William Shakespeare 's Hamlet .
The Lion King; The Lion King tells the story of Simba , a young lion who is to succeed his father , Mufasa , as King of the Pride Lands ; however , after Simba 's uncle Scar murders Mufasa , Simba is manipulated into thinking he was responsible and flees into exile .
**Original claim:** The Lion King is about lions.
**Mutated claim:** The Lion King has nothing to do with lions.

**Evidence:** Indiana Jones; Henry Walton " Indiana " Jones Jr. ( also shortened to Indy ) is a fictional character and the protagonist of the Indiana Jones franchise .
Indiana Jones; George Lucas created the character in homage to the action heroes of 1930s film serials .
**Original claim:** Indiana Jones is fictional.
**Mutated claim:** Indiana Jones is real.

**Evidence:** Scott Eastwood; Scott Eastwood ( born Scott Clinton Reeves ; March 21 , 1986 ) is an American actor , model , and professional skydiver .
Scott Eastwood; He has also been the model for the fragrance Cool Water by Davidoff .
**Original claim:** Scott Eastwood worked as a model.
**Mutated claim:** Scott Eastwood was incapable of working as a model.

**Evidence:** Akshay Kumar; Kumar is also a producer and martial artist who has appeared in over a hundred Hindi films .
Akshay Kumar; Having done so , he has established himself as a leading contemporary actor of Hindi cinema .
**Original claim:** Akshay Kumar works in Hindi cinema.
**Mutated claim:** Akshay Kumar does not work in Hindi cinema.

**Evidence:** Gorillaz; Gorillaz are an English virtual band created in 1998 by musician Damon Albarn and artist Jamie Hewlett .
Virtual band; In music , a virtual band ( also called a virtual group , cartoon group , or cartoon band ) is any group whose members are not corporeal musicians , but animated characters .
**Original claim:** Gorillaz is a British virtual band.
**Mutated claim:** Gorillaz is a German live band.

**Evidence:** Grant Gustin; Thomas Grant Gustin ( born January 14 , 1990 ) is an American actor , singer , and dancer .
Grant Gustin; He is known for his role as Barry Allen / the Flash ( based on the DC Comics character of the same name ) on the CW series The Flash and Arrow , both in the Arrowverse television franchise , and his role as Sebastian Smythe on the Fox series Glee .
**Original claim:** Grant Gustin is a singer.
**Mutated claim:** Grant Gustin is only a writer.

**Evidence:** RB Leipzig; RasenBallsport Leipzig e.V. , commonly known as RB Leipzig , is a German association football club based in Leipzig , Saxony .
Football in Germany; Football is the most popular sport in Germany .
**Original claim:** RB Leipzig plays the most popular German sport.
**Mutated claim:** RB Leipzig plays the least popular German sport.

**Evidence:** One World Trade Center; One World Trade Center ( also known as 1 World Trade Center , 1 WTC or Freedom Tower ) is the main building of the rebuilt World Trade Center complex in Lower Manhattan , New York City .
World Trade Center (2001–present); The original World Trade Center featured the landmark Twin Towers , which opened in 1973 , and were the tallest buildings in the world at their completion .
**Original claim:**  One World Trade Center opened in 2014.
**Mutated claim:** One World Trade Center opened in 1876.

Table 7: Few-shot exemplars prompt of the *negate* mutation type for the FEI task.

**Evidence:** Notes on a Scandal (film); The soundtrack was composed by Philip Glass .
Philip Glass; Philip Morris Glass ( born January 31 , 1937 ) is an American composer .
**Original claim:** Notes on a Scandal has a soundtrack composed by an American.
**Mutated claim:** Notes on a Scandal has a soundtrack composed by an Armenian.

**Evidence:** The Lion King; The Lion King is a 1994 American animated epic musical film , produced by Walt Disney Feature Animation and released by Walt Disney Pictures .
The Lion King; It is the 32nd Disney animated feature film .
**Original claim:** The Lion King is a film.
**Mutated claim:** The Lion King is a TV show.

**Evidence:** Dead Man Down; Dead Man Down is an 2013 American neo-noir crime thriller film written by J.H. Wyman and directed by Danish director Niels Arden Oplev .
Dead Man Down; The film stars Colin Farrell , Noomi Rapace , Dominic Cooper , and Terrence Howard , and was released on March 8 , 2013 .
**Original claim:** Dead Man Down was released in 2013.
**Mutated claim:** Dead Man Down was released in 2014.

**Evidence:** Nick Jonas (album); It was released on November 10 , 2014 , by Island Records .
Island Records; Island Records is a British-American record label that operates as a division of Universal Music Group ( UMG ) .
**Original claim:** Nick Jonas was released by a British-American record label.
**Mutated claim:** Nick Jonas was released by a Chinese-Mongolian record label.

**Evidence:** Deadpool; Created by artist/writer Rob Liefeld and writer Fabian Nicieza , the character first appeared in The New Mutants # 98 ( cover-dated February 1991 ) .
Deadpool; Initially Deadpool was depicted as a supervillain when he made his first appearance in The New Mutants and later in issues of X-Force , but later evolved into his more recognizable antiheroic persona .
**Original claim:** Deadpool first appeared in The New Mutants.
**Mutated claim:** Deadpool first appeared in X-Men.

**Evidence:** Blue-ringed octopus; The blue-ringed octopodes ( genus Hapalochlaena ) are three octopus species that live in tide pools and coral reefs in the Pacific and Indian Oceans , from Japan to Australia .
Blue-ringed octopus; They are recognized as one of the world 's most venomous marine animals .
**Original claim:** The blue-ringed octopus is a marine animal.
**Mutated claim:** The blue-ringed octopus is a terrestrial animal.

**Evidence:** Room (2015 film); Room is a 2015 independent drama film directed by Lenny Abrahamson and written by Emma Donoghue , based on her novel of the same name .
Room (novel); Room is a 2010 novel by Irish-Canadian author Emma Donoghue .
**Original claim:** Room is based on a novel of the same name.
**Mutated claim:** Room is based on a short story of the same name.

**Evidence:** Steve Buscemi; He made his directorial debut in 1996 , with Trees Lounge , in which he also starred .
Trees Lounge; Trees Lounge is a 1996 feature film and the debut of Steve Buscemi as writer and director .
**Original claim:** Steve Buscemi directed the film Trees Lounge.
**Mutated claim:** Steve Buscemi directed the television show Trees Lounge.

Table 8: Few-shot exemplars prompt of the *substitute similar* mutation type for the FEI task.

**Evidence:** Kurt Angle; He then won a freestyle wrestling gold medal at the 1996 Summer Olympics .
Kurt Angle; After graduating college , Angle won a gold medal in freestyle wrestling at the 1995 World Wrestling Championships .
**Original claim:** Kurt Angle is a professional wrestler.
**Mutated claim:** Kurt Angle is a fish.

**Evidence:** Selena Gomez; Between 2009 and 2011 , Gomez starred in films such as Princess Protection Program , Ramona and Beezus , and Monte Carlo , and took on a more mature role in Spring Breakers ( 2013 ) .
Princess Protection Program; Princess Protection Program is a 2009 Disney Channel Original Movie , directed by Allison Liddi-Brown and starring Demi Lovato and Selena Gomez .
**Original claim:** Selena Gomez starred in Princess Protection Program.
**Mutated claim:** Selena Gomez reviewed Princess Protection Program.

**Evidence:** Sterling Archer; Sterling Malory Archer , known simply as Archer , is the titular character and the main protagonist of the American animated comedy series Archer .
Archer (TV series); Archer is an American adult animated spy sitcom created by Adam Reed for the FX network .
**Original claim:** Sterling Archer is the main character of a comedy series.
**Mutated claim:** Sterling Archer directed a comedy series.

**Evidence:** United States Congress; The House of Representatives has six non-voting members in addition to its 435 voting members .
United States Congress; Congress has 535 voting members : 435 Representatives and 100 Senators .
**Original claim:** The United States Congress has 435 Representatives.
**Mutated claim:** The United States Congress has 435 minefields.

**Evidence:** Carole King; Carole King ( born Carol Joan Klein , February 9 , 1942 ) is an American composer and singer-songwriter .
Carole King; She is the most successful female songwriter of the latter half of the 20th century , having written or co-written 118 pop hits on the Billboard Hot 100 between 1955 and 1999 .
**Original claim:** Carole King is an American.
**Mutated claim:** Carole King is an acrobat.

**Evidence:** Manatee; Manatees ( family Trichechidae , genus Trichechus ) are large , fully aquatic , mostly herbivorous marine mammals sometimes known as sea cows .
Herbivore; A herbivore is an animal anatomically and physiologically adapted to eating plant material , for example foliage , for the main component of its diet .
**Original claim:** Manatees are similar to cows on land.
**Mutated claim:** Manatees eat cows on land.

**Evidence:** Lion (2016 film); Lion is a 2016 biographical film directed by Garth Davis ( in his feature debut ) and written by Luke Davies , based on the non-fiction book A Long Way Home by Saroo Brierley with Larry Buttrose .
Garth Davis; Garth Davis is an Australian television , film and commercial director , best known for directing episodes of the series Top of the Lake ( 2013 ) , for which he received Emmy and BAFTA nominations .
**Original claim:** Lion was directed by Garth Davis.
**Mutated claim:** Lion was directed by plants.

**Evidence:** Tropic Thunder; Tropic Thunder is a 2008 satirical action comedy film co-written , produced , and directed by Ben Stiller .
Tropic Thunder; It was written by Stiller , Justin Theroux and Etan Cohen .
**Original claim:** Tropic Thunder was written by Justin Theroux.
**Mutated claim:** Tropic Thunder was awoken by Justin Theroux.

Table 9: Few-shot exemplars prompt of the *substitute dissimilar* mutation type for the FEI task.

**Evidence:** Bryan Adams; He has also won MTV , ASCAP , American Music awards , three Ivor Novello Awards for song composition and has been nominated five times for Golden Globe Awards and three times for Academy Awards for his songwriting for films .
Ivor Novello Awards; The Ivor Novello Awards , named after the Cardiff-born entertainer Ivor Novello , are awards for songwriting and composing .
**Original claim:** Bryan Adams has won Ivor Novello Awards.
**Mutated claim:** Bryan Adams has won Ivor Novello Awards for his singing.

**Evidence:** Malcolm Young; Malcolm Mitchell Young ( born 6 January 1953 ) is an Australian retired musician and songwriter , best known as a co-founder , rhythm guitarist , backing vocalist and songwriter for the hard rock band AC/DC . Malcolm Young; Except for a brief absence in 1988 , he was with the band from its November 1973 beginning until retiring permanently in 2014 , due to health reasons .
**Original claim:** Malcolm Young co-founded AC/DC in 1973.
**Mutated claim:** Malcolm Young co-founded AC/DC in July 1973.

**Evidence:** Gillian Jacobs; Jacobs has also had a recurring role as Mimi-Rose Howard on the HBO series Girls and has appeared in films such as Gardens of the Night ( 2008 ) , The Lookalike ( 2014 ) , Life Partners ( 2014 ) , Hot Tub Time Machine 2 ( 2015 ) , Do n't Think Twice ( 2016 ) and Brother Nature ( 2016 ) .
Hot Tub Time Machine 2; Hot Tub Time Machine 2 is a 2015 American comedy film directed by Steve Pink and written by Josh Heald .
**Original claim:** Gillian Jacobs appeared in the film Hot Tub Time Machine 2.
**Mutated claim:** Gillian Jacobs appeared in the horror film Hot Tub Time Machine 2.

**Evidence:** Marc Maron; He has been host of The Marc Maron Show and co-host of both Morning Sedition and Breakroom Live , all politically oriented shows produced by Air America Media .
The Marc Maron Show; It featured interviews ( both political and showbusiness ) , live comedy , and extensive banter between Maron and Jim Earl , Maron 's co-host , who provides humorous introductions after each commercial break and plays several of the recurring characters in the show 's skits .
**Original claim:** Marc Maron was the host of The Marc Maron Show.
**Mutated claim:** Marc Maron was the only host of The Marc Maron Show.

**Evidence:** The Offspring; The band 's third studio album , Smash ( 1994 ) , became their first commercial success , and has sold over eleven million copies worldwide , setting a record for most albums sold on an independent label and becoming the first album on Epitaph to obtain gold and platinum status .
Smash (The Offspring album); Recording and production were finished a month later , and the album was released on April 8 , 1994 on Epitaph Records .
**Original claim:** The Offspring released Smash in 1994.
**Mutated claim:** The Offspring released Smash in May of 1994.

**Evidence:** NASA; Since that time , most US space exploration efforts have been led by NASA , including the Apollo Moon landing missions , the Skylab space station , and later the Space Shuttle .
Skylab; Skylab was the United States ' first space station , orbiting Earth from 1973 to 1979 , when it fell back to Earth amid huge worldwide media attention .
**Original claim:** NASA is responsible for the Skylab space station.
**Mutated claim:** NASA is responsible for the Skylab space station, launched in 1980.

**Evidence:** Golden State Warriors; The Warriors compete in the National Basketball Association ( NBA ) as a member of the league 's Western Conference Pacific Division .
National Basketball Association; It has 30 teams ( 29 in the United States and 1 in Canada ) , and is an active member of USA Basketball ( USAB ) , which is recognized by FIBA ( also known as the International Basketball Federation ) as the national governing body for basketball in the United States .
**Original claim:** The Golden State Warriors are in the NBA.
**Mutated claim:** The Golden State Warriors are one of 32 teams in the NBA.

**Evidence:** Morrissey; Born in Davyhulme , Lancashire , to a working-class Irish migrant family , Morrissey grew up in Manchester .
Morrissey; Steven Patrick Morrissey ( born 22 May 1959 ) , professionally known as Morrissey , is an English singer , songwriter and author .
**Original claim:** Morrissey was born into a working-class family.
**Mutated claim:** Morrissey was born into a working-class family in 1983.

Table 10: Few-shot exemplars prompt of the *specific* mutation type for the FEI task.