# OpenReview forum: "PivotFEC: Enhancing Few-shot Factual Error Correction with a Pivot Task Approach using Large Language Models"
_EMNLP/2023/Conference — EMNLP 2023 Findings_

### Official Review · Reviewer_8rV1 · 2023-08-03

**Soundness:** 3

**Excitement:**

2: Mediocre: This paper makes marginal contributions (vs non-contemporaneous work), so I would rather not see it in the conference.

**Paper Topic And Main Contributions:**

#############

Update: The authors have conducted experiments to address the reviewer's concern and the results and justification seem convincing. The reviewer would like to raise the soundness score from 2 to 3.

#############

This paper presents PivotFEC, a method for improving factual error correction (FEC) through a “so-called novel” pivot task named factual error injection with the help of LLMs. The authors leverage the capabilities of LLMs in few-shot learning to generate an abundance of paired data, overcoming the scarcity of such data. The method outperforms existing distantly supervised models on the FECDATA dataset, achieving new SOTA result.

**Reasons To Accept:**

1. The proposed pipeline is easy to understand.
2. The experiments consist of reasonable evaluations and include necessary baselines.

**Reasons To Reject:**

1. The major fous of this work is proposing a pivot task “factual error injection”(FEI) that is helpful for enhancing factual error correction. Basically the authors will train a factual error corrector with synthetic data generated from LLM via FEI prompting. **However, the reviewer does not find the novelty of this proposed method** as this method does not differ from previous methods. In lines 190-195, the authors mentioned that previous methods need to do the correction by following (1) first identify factual errors within false claims; and (2) mask these errors with the FVC before using the corrector to revise them, where incorrect masking may cause a model to not perform well. However, one can use methods such as mask-and-filling to create mutated claim like what the authors want to create. For example, just randomly mask out spans (can be noun phrase, verb, etc) and then ask a model to predict the mask (see https://arxiv.org/pdf/2109.09209.pdf section 5.2 for example). One can choose words from the bottom of the beam to increase the chance that a mutated claim is wrong. In this way, in the example from figure 1: “The 2013 NBA draft was held in Los Angeles.” one can just mask “Los Angeles” or “2013” or add an extra mask before the word “held” to create all three types of wrong claims mentioned in the figure and there is even no need to refer to any evidence.

2. Given that this work involves LLMs, **the reviewer has doubt on the quality of the dataset being evaluated on**. It seems like the errors from this evaluation dataset is quite obvious, such as adding negations, substitute dissimilar, etc. **Do LLMs or most recent models (the reviewer means non-LLM based here) really make these types of errors? If not, what is the use case of identifying these errors well?** It’s more interesting to see experiments with datasets that really consist of errors recent models make. Re: “Obvious errors”: the reviewer is not saying these errors are easy to detect but instead emphasize that it is quite easy to construct such synthetic data which actually current LLMs may not make at all.

**Reproducibility:**

4: Could mostly reproduce the results, but there may be some variation because of sample variance or minor variations in their interpretation of the protocol or method.

**Reviewer Confidence:**

4: Quite sure. I tried to check the important points carefully. It's unlikely, though conceivable, that I missed something that should affect my ratings.

---

> ### Author Rebuttal · Authors · 2023-08-29
>
> Thank you for your thoughtful and valuable comments
>
> > *Q1*: Differences between our proposed method and previous methods
>
> Given the absence of paired data (i.e., false claims and their corresponding correct claims), previous works have commonly resorted to the 'mask-then-correct' paradigm.
> This paradigm relies solely on unpaired false claims and correct claims, thus being called  distantly supervised methods.
> Our proposed method differs from previous works in two aspects:
>
> (1) Previous methods do not train correctors on synthetic data for factual error correction. They follow the 'mask-then-correct' pipeline to revise false claims.
> Consider the false claim ''The 2013 NBA draft was held in Pennsylvani'' (Refer to Figure 1 in our paper for more details). For instance,
> they first use a masker to identify the factual errors and then mask them. After these operations, one may get a masked claim ''The 2013 NBA draft was held in [MASK]." Finally, they will feed the masked claim into the corrector, and expect the corrector to generate correct claims (The corrector is trained to reconstruct correct claims conditioned on masked claims and evidence).
>
>
> However, these methods require a masker to explicitly identify factual errors within false claims before revising with a corrector.
> Given the absence of paired data to train the masker, accurately pinpointing factual errors within claims is non-trivial.
> Consequently, the performance of prior works suffers due to this bottleneck.
>
> (2) In this work, we propose to use LLMs to deliberately inject factual errors into correct claims. By doing so, we can obtain a substantial set of synthetic paired data, which will be used to train the corrector for factual error correction.
> Our proposed corrector operates without necessitating a masker, effectively bypassing the bottleneck associated with the explicit identification of factual errors.
>
> > *Q2*: Comparison between LLMs and rule-based methods for constructing datasets
>
> To address the concerns raised, we implement two rule-based methods to generate synthetic paired data, and then train two factual error correctors on them. These two rule-based methods are used to create synthetic data for improving the faithfulness in text summarization (refer to https://arxiv.org/pdf/2109.09209.pdf for more details).
> \
> (1) We first resort to spacy, a free open-source library for NLP, to recognize name entities in correct claims.
> We then create false claims by swapping named entities from the correct claims with alternative entities of the same entity type randomly chosen from the training dataset. This method is referred to as 'SwapEntity' (this method is mentioned in section 5.1 in https://arxiv.org/pdf/2109.09209.pdf).
> \
> (2) The second method resorts to the mask-then-fill pipeline to create false claims.
>  In this approach, named entities within the correct claims are replaced with [MASK] tokens. The masked claims are then processed through the BART model to generate a new claim by filling the [MASK] tokens. These newly generated claims are considered as false claims. This method is termed 'MaskEntity' (this method is mentioned in section 5.2 in https://arxiv.org/pdf/2109.09209.pdf).
>
> Since the other two methods mentioned in sections 5.3 and 5.4 in the paper about faithful summarization (https://arxiv.org/pdf/2109.09209.pdf) rely on paired data, they are not applicable to this task. Both our work and previous works try to solve factual error corrections with the assumption that paired data is not available.
>
> The experimental results are summarized in the subsequent table. Our proposed model, 'PivotFEC', leverages LLMs (in this table, using ChatGPT) to inject factual errors into correct claims. Results in this table indicate that our proposed method notably outperforms the rule-based methods.
> Furthermore, our proposed model achieves the peak performance with just 2000 synthetic data instances for training the corrector, whereas rule-based methods necessitate 10000 synthetic data instances to reach their peak performances for corrector training.
> These results can be attributed to the enhanced quality of false claims produced by ChatGPT. Rule-based methods, by contrast, often produce false claims with suboptimal quality in two key aspects: 1) Grammatical errors might be present in rule-based generated false claims. 2) false claims generated by rule-based methods might deviate from the original topics present in correct claims.
> Given the remarkable in-context learning capabilities of LLMs, factual errors can be introduced without introducing grammatical errors and deviating from the original topics.
>
> We will add these rule-based methods and their performances to the final version of our paper.
>
> | Models/Metrics | SARI Final | ROUGE-2 |
> | ------------- | ------------- | ------------- |
> | PivotFEC  | **66.30**  | **66.68** |
> | SwapEntity  | 59.71  |62.06 |
> |MaskEntity |61.24 |63.49 |
>
>
>
> > *Q3*: The necessity of evidence
>
> As shown in the second table, the mutated claim generated by MaskEntity with BART without using evidence (in this case, we use bart-base)  deviates from the original topic. By comparison, the mutated claim generated by ChatGPT with evidence (our proposed method) does not deviate from the original topic. Evidence can avoid arbitrary generation when models generate false claims, thus it is helpful to keep topics in correct claims.
>
> |     |   |
> |  ----  | ----  |
> |Evidence| Julianne Moore; From 1985 to 1988, she was a regular in the soap opera As the World Turns, earning a Daytime Emmy for her performance.|
> | Original claim  |Julianne Moore was in the television series As the World Turns.|
> | Masked claim  | \<mask\> was in the television series As the World Turns. |
> | Mutated claim generated by MaskEntity | The first time I saw him was in the television series As the World Turns. |
> | Mutated claim generated by ChatGPT | Julianne Moore never appeared in any television series.|
> |     |   |
>
>
> > *Q4*: The quality of the dataset being evaluated on
>
> The dataset used in this paper is extracted from the well-known fact verification dataset, FEVER (https://aclanthology.org/N18-1074/). The authors asked annotators to introduce factual errors into correct claims with six types of mutation: paraphrasing, negation, substitution of anentity/relation with a similar/dissimilar one, and making the claim more general/specific.
> The two most recent works have employed this dataset to evaluate factual error correction systems, and hence, we align with their settings.
>
>
> On the other hand, to evaluate on data generated by LLMs, one needs to gather text generated by LLMs.
> However, for the purpose of automatic evaluation, one also requires a reference "gold standard" correction.
> Thus, one needs annotators to rectify factual errors in the text generated by LLMs to obtain accurate references. As far as we are aware, such a dataset is currently unavailable. Nevertheless, our focus is not on introducing a new dataset with this work. Therefore, we will leave this as a task for future endeavors.
>
> Ultimately, within the table provided below, we showcase several factual errors originating from ChatGPT. These errors are related  to the concept of 'substitute similar' factual errors discussed in our paper.
>
>
> | Input  to ChatGPT| Output generated by ChatGPT | The true fact |
> | ------------- | ------------- | ------------- |
> | Please answer the question in one sentence: Which team won the 2022 NBA champion？  | The Phoenix Suns won the 2022 NBA championship. | Golden State Warriors won the 2022 NBA championship. |
> | Did Stefen curry won the NBA Finals MVP in 2015?  | Yes, Stephen Curry won the NBA Finals MVP in 2015. He played a significant role in leading the Golden State Warriors to the NBA championship that year. Curry's exceptional shooting skills and overall performance during the finals earned him the honor of being named the Finals MVP.  | Andre Iguodala won the NBA Finals MVP in 2015. |

---

### Official Review · Reviewer_PYxa · 2023-08-05

**Soundness:** 4

**Excitement:**

3: Ambivalent: It has merits (e.g., it reports state-of-the-art results, the idea is nice), but there are key weaknesses (e.g., it describes incremental work), and it can significantly benefit from another round of revision. However, I won't object to accepting it if my co-reviewers champion it.

**Paper Topic And Main Contributions:**

This paper attempts to address the task of factual error correction, where there is a lack of data that contains false claims. By generating synthetic data using llms and creating the pivot task of factual error injection, the proposed method outperforms existing distantly
supervised methods and few-shot methods.

**Reasons To Accept:**

- The paper is clearly written and easy to follow.
- The paper contains thorough experiments, including human evaluation, that show the strength of the proposed method as well as its weaknesses.

**Reasons To Reject:**

1. Unless I missed it, but I don't see the need to use llm for generating false claims. In summarization, many works on improving factuality/faithfulness of the model and developing factuality detection metrics, have generated synthetic data (of unfaithful summaries) using simpler rule-based methods that are effective. This may also reduces the potential noise from ChatGPT as discussed in section 5.3. I am wondering how reliable chatgpt is in terms of reproducibility (given its constant updates). Some qualitative analysis would be nice on this part.

**Reproducibility:**

4: Could mostly reproduce the results, but there may be some variation because of sample variance or minor variations in their interpretation of the protocol or method.

**Reviewer Confidence:**

3: Pretty sure, but there's a chance I missed something. Although I have a good feel for this area in general, I did not carefully check the paper's details, e.g., the math, experimental design, or novelty.

---

> ### Author Rebuttal · Authors · 2023-08-29
>
> Thank you for your thoughtful and valuable feedback.
>
> > *Q1*: Comparison between LLMs and rule-based methods for constructing datasets
>
> To address the concerns raised, we implement two rule-based methods to generate synthetic paired data, and then train two factual error correctors on them. These two rule-based methods are used to create synthetic data for improving the faithfulness in text summarization (refer to https://arxiv.org/pdf/2109.09209.pdf for more details).
> \
> (1) We first resort to spacy, a free open-source library for NLP, to recognize name entities in correct claims.
> We then create false claims by swapping named entities from the correct claims with alternative entities of the same entity type randomly chosen from the training dataset. This method is referred to as 'SwapEntity' (this method is mentioned in section 5.1 in https://arxiv.org/pdf/2109.09209.pdf).
> \
> (2) The second method resorts to the mask-then-fill pipeline to create false claims.
>  In this approach, named entities within the correct claims are replaced with [MASK] tokens. The masked claims are then processed through the BART model to generate a new claim by filling the [MASK] tokens. These newly generated claims are considered as false claims. This method is termed 'MaskEntity' (this method is mentioned in section 5.2 in https://arxiv.org/pdf/2109.09209.pdf).
>
> Since the other two methods mentioned in sections 5.3 and 5.4 in the paper about faithful summarization (https://arxiv.org/pdf/2109.09209.pdf) rely on paired data, they are not applicable to this task. Both our work and previous works try to solve factual error corrections with the assumption that paired data is not available.
>
> The experimental results are summarized in the subsequent table. Our proposed model, 'PivotFEC', leverages LLMs (in this table, using ChatGPT) to inject factual errors into correct claims. Results in this table indicate that our proposed method notably outperforms the rule-based methods.
> Furthermore, our proposed model achieves the peak performance with just 2000 synthetic data instances for training the corrector, whereas rule-based methods necessitate 10000 synthetic data instances to reach their peak performances for corrector training.
> These results can be attributed to the enhanced quality of false claims produced by ChatGPT. Rule-based methods, by contrast, often produce false claims with suboptimal quality in two key aspects: 1) Grammatical errors might be present in rule-based generated false claims. 2) false claims generated by rule-based methods might deviate from the original topics present in correct claims.
> Given the remarkable in-context learning capabilities of LLMs, factual errors can be introduced without introducing grammatical errors and deviating from the original topics.
>
> We will add these rule-based methods and their performances to the final version of our paper.
>
> | Models/Metrics | SARI Final | ROUGE-2 |
> | ------------- | ------------- | ------------- |
> | PivotFEC  | **66.30**  | **66.68** |
> | SwapEntity  | 59.71  |62.06 |
> |MaskEntity |61.24 |63.49 |
>
>
> > *Q2*: Reproducibility
>
> As per the OpenAI documentation available at https://platform.openai.com/docs/models/gpt-3-5, continuous updates are being made to enhance ChatGPT (referred to as gpt-3.5-turbo). However, OpenAI also maintains the snapshots of the gpt-3.5-turbo model for a certain period. In our study, we have employed gpt-3.5-turbo-0301, which captures the state of the gpt-3.5-turbo model from March 1st, 2023. This snapshot will remain accessible until June 13th, 2024, thereby ensuring the reproducibility of our research.

---

### Official Review · Reviewer_Ejt3 · 2023-08-05

**Soundness:** 3

**Excitement:**

4: Strong: This paper deepens the understanding of some phenomenon or lowers the barriers to an existing research direction.

**Paper Topic And Main Contributions:**

The paper presents a pivot task, factual error injection that makes factual error to generate false claim. To generate false claim, the paper leverages large language models like ChatGPT by using few shot examples to the prompt. This claim is used to train a factual error correction system. Experimental results in the paper show that the factual error corrector trained with the generated dataset outputperform the original counterpart (e.g. ChatGPT).

**Reasons To Accept:**

- The paper introduces an interesting and an effective way to generate false claim that can be used to train a factual error correction system.
- Proposed approach is effective and achieves a new state-of-the-arts in factual error correction task.
- The paper provides decent analysis on the proposed approach such as the effect of number of in-context examples and the size of LLMs.

**Reasons To Reject:**

- Although the generated false claims using LLMs are shown to be helpful in training correction system, I have some doubts about the key difference between previous methods to generate synthetic false claims such as mask infilling or entity substitution in the previous papers. Why LLMs are more helpful in generating pivot FEC task (False Claim Injection) ? Are there any evidences that the generated false claim can better model the false claims in the real world (or the model generated claim)?

**Reproducibility:**

4: Could mostly reproduce the results, but there may be some variation because of sample variance or minor variations in their interpretation of the protocol or method.

**Reviewer Confidence:**

3: Pretty sure, but there's a chance I missed something. Although I have a good feel for this area in general, I did not carefully check the paper's details, e.g., the math, experimental design, or novelty.

---

> ### Author Rebuttal · Authors · 2023-08-29
>
> Thanks for your thoughtful and valuable comments.
>
> > *Q1*: Differences between our proposed method and previous methods
>
> Given the absence of paired data (i.e., false claims and their corresponding correct claims), previous works have commonly resorted to the 'mask-then-correct' paradigm.
> This paradigm relies solely on unpaired false claims and correct claims, thus being called  distantly supervised methods.
> Our proposed method differs from previous works in two aspects:
>
> (1) Previous methods do not train correctors on synthetic data for factual error correction. They follow the 'mask-then-correct' pipeline to revise false claims.
> Consider the false claim ''The 2013 NBA draft was held in Pennsylvani'' (Refer to Figure 1 in our paper for more details). For instance,
> they first use a masker to identify the factual errors and then mask them. After these operations, one may get a masked claim ''The 2013 NBA draft was held in [MASK]." Finally, they will feed the masked claim into the corrector, and expect the corrector to generate correct claims (The corrector is trained to reconstruct correct claims conditioned on masked claims and evidence).
>
>
> However, these methods require a masker to explicitly identify factual errors within false claims before revising with a corrector.
> Given the absence of paired data to train the masker, accurately pinpointing factual errors within claims is non-trivial.
> Consequently, the performance of prior works suffers due to this bottleneck.
>
> (2) In this work, we propose to use LLMs to deliberately inject factual errors into correct claims. By doing so, we can obtain a substantial set of synthetic paired data, which will be used to train the corrector for factual error correction.
> Our proposed corrector operates without necessitating a masker, effectively bypassing the bottleneck associated with the explicit identification of factual errors.
>
> > *Q2*: Comparison between LLMs and rule-based methods for constructing datasets
>
> To address the concerns raised, we implement two rule-based methods to generate synthetic paired data, and then train two factual error correctors on them. These two rule-based methods are used to create synthetic data for improving the faithfulness in text summarization (refer to https://arxiv.org/pdf/2109.09209.pdf for more details).
> \
> (1) We first resort to spacy, a free open-source library for NLP, to recognize name entities in correct claims.
> We then create false claims by swapping named entities from the correct claims with alternative entities of the same entity type randomly chosen from the training dataset. This method is referred to as 'SwapEntity' (this method is mentioned in section 5.1 in https://arxiv.org/pdf/2109.09209.pdf).
> \
> (2) The second method resorts to the mask-then-fill pipeline to create false claims.
>  In this approach, named entities within the correct claims are replaced with [MASK] tokens. The masked claims are then processed through the BART model to generate a new claim by filling the [MASK] tokens. These newly generated claims are considered as false claims. This method is termed 'MaskEntity' (this method is mentioned in section 5.2 in https://arxiv.org/pdf/2109.09209.pdf).
>
> Since the other two methods mentioned in sections 5.3 and 5.4 in the paper about faithful summarization (https://arxiv.org/pdf/2109.09209.pdf) rely on paired data, they are not applicable to this task. Both our work and previous works try to solve factual error corrections with the assumption that paired data is not available.
>
> The experimental results are summarized in the subsequent table. Our proposed model, 'PivotFEC', leverages LLMs (in this table, using ChatGPT) to inject factual errors into correct claims. Results in this table indicate that our proposed method notably outperforms the rule-based methods.
> Furthermore, our proposed model achieves the peak performance with just 2000 synthetic data instances for training the corrector, whereas rule-based methods necessitate 10000 synthetic data instances to reach their peak performances for corrector training.
> These results can be attributed to the enhanced quality of false claims produced by ChatGPT. Rule-based methods, by contrast, often produce false claims with suboptimal quality in two key aspects: 1) Grammatical errors might be present in rule-based generated false claims. 2) false claims generated by rule-based methods might deviate from the original topics present in correct claims.
> Given the remarkable in-context learning capabilities of LLMs, factual errors can be introduced without introducing grammatical errors and deviating from the original topics.
>
> We will add these rule-based methods and their performances to the final version of our paper.
>
> | Models/Metrics | SARI Final | ROUGE-2 |
> | ------------- | ------------- | ------------- |
> | PivotFEC  | **66.30**  | **66.68** |
> | SwapEntity  | 59.71  |62.06 |
> |MaskEntity |61.24 |63.49 |

---

### Meta-Review · Area_Chair_fGtj · 2023-09-19

**Recommendation:** 2

**Metareview:**

The paper studies the task of factual error correction - recovery of a true claim from a misinformed claim by revising the sentence to be factually accurate. The paper contributes a method for generating synthetic false claims which aid in training a factual error corrections system.

The paper claims to aid training data sparsity by introducing factual error injection as a task to go generate the synthetic data. This approach seems well motivated. But I would also suggest that the authors compare their work to Cao et al 2020 https://aclanthology.org/2020.emnlp-main.506.pdf in the literature survey who perform artificial corruptions of summaries to train an error correction system. Much like this paper. There is value in the author's paper, especially relying on an LLM rather than a set of text transformations.

There is reasonable consistency among the reviews for this paper, scoring 3.3 for soundness and 3 for excitement. I think the experiments provided show the the value of the methods proposed by the authors. There could be some scope for error detection and correction from LLMs in a larger context.  The authors rebutted the reviews well. However there reviewers are still concerns about novelty, and quality of generated instances that could be further addressed.

Regarding novelty, the use of the LM as a corruptor is interesting, but other methods for data corruption injection have been identified (e.g. the contrastive method of CLIFF, or the distantly supervised method of cao et al., 2020).  Comparing LLM-generated corruptions might be of interest to the community so I am not sure about the strength of this negative

Regarding quality, I would also encourage the authors to manually verify the generated instances to better understand the fluency and specificity requirements suggested by T+V2021.

---

### Decision · Program_Chairs · 2023-10-07

**Decision:**

Accept-Findings

**Comment:**

The paper studies the task of factual error correction - recovery of a true claim from a misinformed claim by revising the sentence to be factually accurate. The paper contributes a method for generating synthetic false claims which aid in training a factual error corrections system.

The paper claims to aid training data sparsity by introducing factual error injection as a task to go generate the synthetic data. This approach seems well motivated. But I would also suggest that the authors compare their work to Cao et al 2020 https://aclanthology.org/2020.emnlp-main.506.pdf in the literature survey who perform artificial corruptions of summaries to train an error correction system. Much like this paper. There is value in the author's paper, especially relying on an LLM rather than a set of text transformations.

There is reasonable consistency among the reviews for this paper, scoring 3.3 for soundness and 3 for excitement. I think the experiments provided show the the value of the methods proposed by the authors. There could be some scope for error detection and correction from LLMs in a larger context.  The authors rebutted the reviews well. However there reviewers are still concerns about novelty, and quality of generated instances that could be further addressed.

Regarding novelty, the use of the LM as a corruptor is interesting, but other methods for data corruption injection have been identified (e.g. the contrastive method of CLIFF, or the distantly supervised method of cao et al., 2020).  Comparing LLM-generated corruptions might be of interest to the community so I am not sure about the strength of this negative

Regarding quality, I would also encourage the authors to manually verify the generated instances to better understand the fluency and specificity requirements suggested by T+V2021.